# Advanced Analysis and Validation of a microRNA Signature for Fanconi Anemia

**DOI:** 10.3390/genes15070820

**Published:** 2024-06-21

**Authors:** Enrico Cappelli, Silvia Ravera, Nadia Bertola, Federica Grilli, Margherita Squillario, Stefano Regis, Paolo Degan

**Affiliations:** 1Haematology Unit, IRCCS Istituto Giannina Gaslini, Via Gerolamo Gaslini 5, 16148 Genova, Italy; federicagrilli@gaslini.org; 2Department of Experimental Medicine, University of Genoa, Via De Toni 14, 16132 Genova, Italy; silvia.ravera@unige.it; 3Molecular Pathology Unit, IRCCS Ospedale Policlinico San Martino, L. go R. Benzi 10, 16132 Genoa, Italy; nadia.bertola@hsanmartino.it; 4LISCOMP Laboratory, IRCCS Ospedale Policlinico San Martino, 16132 Genova, Italy; 5Laboratory of Clinical and Experimental Immunology, IRCCS Istituto Giannina Gaslini, Via Gerolamo Gaslini 5, 16148 Genova, Italy; stefanoregis@gaslini.org; 6Mutagenesi e Prevenzione Oncologica Unit—IRCCS Ospedale Policlinico San Martino, L. go R. Benzi 10, 16132 Genoa, Italy; paolo.degan@virgilio.it

**Keywords:** Fanconi anemia, microRNA regulation, bioinformatic analysis, miR-206, DIANA tools, MIENTURNET

## Abstract

Some years ago, we reported the generation of a Fanconi anemia (FA) microRNA signature. This study aims to develop an analytical strategy to select a smaller and more reliable set of molecules that could be tested for potential benefits for the FA phenotype, elucidate its biochemical and molecular mechanisms, address experimental activity, and evaluate its possible impact on FA therapy. In silico analyses of the data obtained in the original study were thoroughly processed and anenrichment analysis was employed to identify the classes of genes that are over-represented in the FA-miRNA population under study. Primary bone marrow mononuclear cells (MNCs) from sixFA patients and sixhealthy donors as control samples were employed in the study. RNAs containing the small RNA fractions were reverse-transcribed and real-time PCR was performed in triplicate using the specific primers. Experiments were performed in triplicate.The in-silico analysis reported six miRNAs as likely contributors to the complex pathological spectrum of FA. Among these, three miRNAs were validated by real-time PCR. Primary bone marrow mononuclear cells (MNCs) reported a significant reduction in the expression level of miRNA-1246 and miRNA-206 in the FA samples in comparison to controls.This study highlights several biochemical pathways as culprits in the phenotypic manifestations and the pathophysiological mechanisms acting in FA. A relatively low number of miRNAs appear involved in all these different phenotypes, demonstrating the extreme plasticity of the gene expression modulation. This study further highlights miR-206 as a pivotal player in regulatory functions and signaling in the bone marrow mesenchymal stem cell (BMSC) process in FA. Due to this evidence, the activity of miR-206 in FA deserves specific experimental scrutiny. The results, here presented, might be relevant in the management of FA.

## 1. Introduction

Fanconi anemia (FA) is a rare autosomal recessive bone marrow(BM) failure syndrome also characterized by developmental delay, physical abnormalities, and anincreased incidence of solid tumors and leukemias [1] that features very a high cancer predisposition. Approximately 20% of the patients develop some type of malignancy, 45% of which are leukemias. Based on the differential expression data analysis of microRNAs in FA versus normal samples [1], we herein report an advanced bioinformatic approach aimed at the selection of a miRNA population that may likely help to define a practical approach to experimental activity on FA and gather simple and reliable hypotheses for laboratory testing.

MicroRNAs are small non-coding RNA molecules of approximately 22 nucleotides in length that have been established as important regulators of biological and pathological processes [2]. It is believed that as much as 30% of human genes are regulated by miRNAs [3]. MicroRNAs regulate messenger RNAs (mRNAs) in post-transcriptional phases by inhibiting their translation or by degrading the mRNA molecule. Furthermore, miRNA expression level fluctuation makes them valuable biomarker candidates for specific diseases [4]. As the amount of data and applications in miRNomics is increasing rapidly, driven by the fast advances in next-generation sequencing (NGS), the informatic tools supporting their analysis are increasingly adequate and implemented. These include the prediction of miRNA targets, enrichment [5], and annotation tools [6]. Many tools that provide enrichment analyses for miRNAs first convert them to their targets and then perform the analysis on the target genes [7,8]. MicroRNA-expression profiling of human tumors has identified signatures associated with diagnosis, progression, staging, prognosis, and response to treatment. Profiling has been exploited to identify miRNA that might represent downstream targets of activated oncogenic pathways or target protein-coding genes involved in cancer. The main mechanism that underlies changes in the function of miRNAs in cancer cells seems to be aberrant gene expression in comparison with the corresponding normal tissues. To date, every type of tumor analyzed by miRNA profiling has shown significantly different miRNA profiles (for mature and/or precursor miRNAs) compared with normal cells from the same tissue [9]. In conclusion, as we already attempted to elucidate the impact of miRNA in FA, this study aimed to define and characterize the original results [1]. In view of the efforts, we dedicated to defining an miRNA population of interest, through in silico selection and molecular validation, we consider the obtained results as promising tools for future investigations in FA pathology.

## 2. Results and Discussion

Since miRNA regulation is a degenerate functional relationship, where one miRNA targets multiple genes and each target gene can be simultaneously targeted by more than one miRNA, it has become indispensable to elaborate in silico tools able to reasonably shorten the list of miRNA-target interactions.This helps to better focus onthe regulatory mechanism orchestrated by different miRNAs in the various cellular processes so to prioritize miRNA-target interactions and maximize the effectiveness of the downstream analysis. To date, no straightforward process has been devised to assistin data analysis procedures. For example, in our original report [1], the 24 miRNAs resulting from the analysis were distributed in upregulated (9) and downregulated (15) miRNAs, associated with 31 and 30 pathways, respectively. Among these, only 14 upregulated and 13 downregulated pathways were independent. The two miRNA populations influence two different gene populations, which overlap only partially. This problem poses different questions regarding the meaning of this data redundancy as well as which procedure might be the most appropriate to adopt for the analysis. There is not even much literature dedicated to discussing what the preferential approaches might be.However, several sophisticated analysis tools have luckily been developed recently, addressing and overcoming several problems. Among the various analysis tools freely available on the web, we found that MIENTURNET was particularly suited for our purposes due to its versatility and ease of use. MIENTURNET is a user-friendly web tool that integrates the miRNA-target enrichment analysis with a network-based visualization and analysis, addressing the prioritization problem of miRNA-target candidate interactions. The procedure embedded in MIENTURNET assumes basic skills in network theory, and statistics and the software donot require bioinformatics expertise.With MIENTURNET, a list of mature miRNAs or genes can be analyzedto infer all possible computational and experimental evidence of their regulation on target genes, or miRNAs, based on a statistical overrepresentation analysis of their specific interactions. These interactions are consequently visualized as a network and analyzed according to their topological features to gain insights into understanding the biological processes underlying the target gene activity, by acquiring information from different data platforms generated by the analysis: KEGG, Reactome, WikiPathways, and Disease Ontology.Theoriginal population of 24 miRNAs associated with the analyzed FA samples [1], selected through MIENTURNET, yielded a panel of 6 miRNAs (Table 1).These miRNAs are associated with the 17 pathways (Table 2). Table 2 reports data consistent with KEGG MIENTURNET analysis. Alternatively, the software also generates information for Reactome, WikiPathways, and Disease Ontology.

To validate the results obtained in silico, we evaluated the expression of the sixselected miRNAs in bone marrow mononucleated cells.

We detected a significant reduction in the expression level of miRNA-1246 and miRNA-206 in FA-A patients compared to HDs (Figure 1).

MiRNA-30b-5p shows a difference in expression between FA patients and HDs but does not reach significance. MiRNA-129-1-3p, miRNA181b-3p, and miRNA-23b-p5 were very poorly expressed and it was not possible to assess differences between FA patients in comparison to HDs.

The six miRNAs identified after the enrichment analysis plus the validation processes finally yielded 17 pathways as main contributors to the complex biochemical background associated with the FA phenotype (Table 2). This analysis identified 128 differentially expressed genes associated with these pathways (Appendix A).We will focus the following discussion on miR-206 and miR-1246, which show the most relevant response in the population analyzed in this study (schematized in Figure 2).

MiR-206 is associated with pathway 5 linked with ECM-receptor interaction and controls several integrin and laminin gene family members (TGA5, ITGA6, LAMA4, LAMB1, and LAMB2), which often display abnormal expression in several types of cancers [10,11]. ITGA5 and ITGA6 play active roles in cytoskeletal organization and cell migration and their differential expression in head and neck squamous cell carcinoma is of prognostic value [12]. Both laminins and integrins are glycoproteins of the extracellular matrix. As components of the basal lamina, these proteins are involved in cell differentiation, migration, and adhesion, and they are linked with dysfunctional processes from the bone marrow, including the insurgence of multiple myeloma, where the close interaction between myeloma cells and the BM microenvironment plays an important role in its progression [13,14]. Laminins and integrins are partners, among several other proteins and factors (IL-6, IL-8, MMP-2, and MMP-9), in the Epithelial–Mesenchymal Transition (EMT), a critical process in embryonic development, in which epithelial cells undergo trans-differentiation [15]. These processes appear under the control of the NF-κB/TNF-α axis and its inhibition reverses the oncogenic signaling involved in breast cancer progression and restores the control of tissue architecture [16]. Furthermore, overexpression of HIF-1α was accompanied by the acquisition of epithelial–mesenchymal transition features, and hypoxia and serum deprivation are associated with the regulation of the glycosylation patterns in the triple-negative breast cancer profile aggressive features [17]. Several elements concerning this pathway have already emerged in experimental research in FA [18,19,20] and its occurrence in this study warrants a closer view.

MiRNA-1246, together with miR30b-5p, regulates the activity of the p53 signaling pathway (Pathway 7) [21,22,23,24]. It is well known that a defective FA pathway strongly impacts HSPC differentiation and survival through activation of the p53/p21 axis [25,26]. Concerning these findings, the p53 pathway is activated in many pathological conditions, and indeed, many of the pathways downstream of number 7 (namely pathways 9, 11, 13, 14, 17: see Appendix A) appear linked to this functional (and pathological) condition with which they share many of the biomolecular mechanisms. A core gene list is shared among many of these pathways (BAX, CASP3, CCND1, CDK4, CDK6, CDKN1A/p21, CDKN2A, MAPK1, MDM2, NRAS, PIK3CB, TP53). Many of the proteins associated with these genes are key factors in intermediary metabolism and their activation is common in the promotion and progression phases of several cancerogenic processes [27,28]. In FA patients, a defective process in hematopoietic stem and progenitor cells (HSPCs) is observed before the onset of clinical BMF, where DNA damage and replicative stress elicit p53 activity that results in a late p21/Cdkn1a-dependent G0/G1 cell-cycle. This exacerbated p53/p21 response to cellular stress and DNA damage accumulation could be a central mechanism for the progressive HSPC depletion in FA patients, and indeed, p53 knockout can rescue the HSPC defects in FA models both in vitro and in vivo [29]. Since p53 pathway signaling appears as a core signaling in FA, we quantified the expression of TP53, p21, and CDK6, which are target genes of different pathways regulated by miRNA-1246 and miRNA-206 [29,30,31]. Although all genes are more expressed in FA patients than in controls, only CDK6 has statistical significance (Figure 3).

MiRNA-1246 and miRNA-206 also act on pathway 4, oocyte meiosis. The cyclins (CDK/CCN) family, the CPBE (cytoplasmic polyadenylation binding element) family, and the Ser/Thr phosphatase (PPP) family, are elements acting with tight interconnections. The key activity of these processes is the dynamic interplay in protein phosphorylation/dephosphorylation, one of the most important regulatory mechanisms controlling all cellular processes. Alterations in the phosphorylation networks have major consequences in promoting several pathological processes and cancer [32]. Polyadenylation by CPEB elements confers stabilization to the CDK/CCN elements [33], which controls cell cycle progression and DNA replication. An unscheduled CDK/CCN activation results in replication stress and DNA damage that may finally lead to genomic instability and carcinogenesis [34,35]. PPPs are signal-transducing enzymes that dephosphorylate cellular phosphoproteins, thus acting as antagonists for the kinases [36]. It is interesting to note that these functions are finely tuned through redox signals. Cyclin and CDK activities appear sensitive to the redox stress related to mitochondrial activity [37], whereas both CPEB and PPP signaling activity is regulated by redox intermediates and hypoxia-inducible factor 1-α (HIF-1α) [38,39]. Since the impact of oxidative stress on FA has been already well discussed [28,40,41,42], the findings reported here might be helpful for the experimental management of FA. MiR-30b-5p in this pathway may act as a tumor suppressor, repressing cell proliferation and the cell cycle [43,44,45]. It is also associated with breast cancer and NSCLC [45,46]. MiR-1246 can influence the stemness and resistance of cancer cells to therapeutics and appears to have a role in NSCLC [21,47]. There might be a substantial superimposition of the functions controlled by pathway 4 (oocyte meiosis) and pathway 11 (the cell cycle), also related to the contribution of miR-206.

Several of the pathways relevant for the functions controlled by miR-206 and miR-1246 are also regulated by miR-30b-5p (pathways 4, 7, 8, 11, 15: Table 2), which is also a major player in pathways 1, 2, 3, and 6. MiR-30b-5p is associated with different aspects of fatty acid metabolism [26,42] and we recently reported how fatty acid metabolism and biosynthesis are significantly altered in FA [30,42]. MiR-30b-5p overexpression could reduce intracellular lipid deposition and regulate intracellular lipid metabolism by targeting PPAR-α and GLUT1, thus triggering mitochondrial oxidative phosphorylation, fatty acidbeta-oxidation and synthesis, glucose uptake, and gluconeogenesis. Also, the mucin-type O-glycan biosynthesis pathway (pathway 2) appears solely associated with hsa-miR-30b-5p, which controls a group of isoenzymes that initiate several glycosylation reactions [48,49,50]. Aberrant expression of glycan has been reported in most hematological malignancies, such as AML, myeloproliferative neoplasms, and multiple myeloma, as well in bone-marrow-associated breast cancer cell dissemination [51,52,53]. The mucin-type O-glycan biosynthesis pathway links glucose dysmetabolism and type 2 diabetes [54], a well-established phenotypic feature in FA disease, associated with redox unbalance and mitochondrial dysfunctionality [49,50,54].

## 3. Conclusions

In conclusion, this study highlights the role of different miRNAs in the pathophysiological manifestations of FA, in particular, miR-206 and miR-1246 seem suitable candidates for future molecular screening studies and experimental studies. 

MiR-206 introduces itself as a true newcomer in FA as a potential player in regulatory functions and signaling in the BMSC process within the BM niche. To date, this miRNA has been viewed principally as a muscle-specific regulator, while it has a much broader spectrum of action [55,56]. The possible involvement of miR-206 in FA should be considered due to some crucial characteristics of the disease and its occurrence. MiR-206 inhibits osteogenic differentiation of bone marrow mesenchymal stem cells responsible for the lineage commitment and differentiation of skeletal stem cells and bone marrow stromal cells [56,57]. MiR-206 plays a critical role in the hematopoietic lineage [58]. MiR-206 inhibits the ability of migration of mesenchymal stem cells, induces cell apoptosis, and alters the mitochondrial membrane potential [59]. MiR-206 expression regulates glutamine metabolism, which is associated with the maintenance of a hypoxic environment in the BMSCs, necessary for their maturation process [41,60,61,62]. 

It is, however, necessary to underline the role of miR-1246 as an important partner in shaping the pathophysiological manifestations of FA. Indeed, miR-1246 is involved in the defective p53 pathway, in the signaling through cell cycle progression by affecting the balance between phosphorylation/dephosphorylation and cyclins and kinase activities, and in the redox unbalance. 

## 4. Materials and Methods

### 4.1. Bioinformatic Approach

DIANA tools (http://diana.imis.athena-innovation.gr/DianaTools/index.php) (accessed on 15 June 2023) [63] provide algorithms, databases, and software for interpreting and archiving data in a systematic framework ranging from the analysis of expression regulation from deep sequencing data, and the annotation of miRNA regulatory elements and targets, to the interpretation of the role ofnon-codingRNAs in various diseases and pathways. A selection of the most relevant miRNAs associated with the DIANA analysis is obtained through the generation of heat maps, which enable the identification of miRNA subclasses or the GO terms that characterize these miRNAs. The original FA-miRNA signature was successively analyzed with MIENTURNET (MIcroRNAEnrichmentTURnedNETwork) (http://userver.bio.uniroma1.it/apps/mienturnet/ accessed on 15 June 2023) [64], a software that offers the possibility to perform statistical and network-based analyses using an effective tool, leading to a more effective prioritization of the miRNA-target interactions. This has the potential to allow researchers with limited computational and informatics experience to gather information exhaustively thanks to the intuitive web interface. The purpose of the use of these different web tools, which are not mutually exclusive but functionally complementary, was to perform functional enrichment analysis to identify classes of genes or proteins that are over-represented in the large set of genes associated with the FA-miRNA population analyzed, and to associate them with the relevant disease phenotypes. Downstream of these analytical steps, we employed several other web tools to producegraphical representations of the gene-associated miRNA relationship to obtaininformation concerning the biochemical mechanisms underlying the intricate complexity of the gene activities addressed by the different miRNAs, as well as to categorize and identify the genes in terms of pathways, biological functions, and their eventual connections and interdependence. To this end, we employed STRING functional protein association networks (https://string-db.org/ accessed on 15 June 2023) and Cytoscape (https://cytoscape.org/ accessed on 15 June 2023), the open-source software platform for visualizing complex networks (Figure 4).

### 4.2. BM Samples

The institutional review board of the “G. Gaslini” Hospital, Genoa, Italy, approved the study, and all the subjects or their legal guardians gave written informed consent to the investigation according to the Declaration of Helsinki. A total of 6 patients with mutation in the FA complementation group Ageneand 6 healthy donors (HDs) wereinvolved in the study. Primary bone marrow mononuclear cells (MNCs) were isolated using Ficoll-Paque Plus (GE Healthcare Biosciences, Piscataway, NJ, USA).

### 4.3. RNA Isolation, Expression of miRNA, and Selected Gene Targets

RNA containing the small RNA fraction was extracted using the RNeasy Plus mini kit (Qiagen, Hilden, Germany) following a dedicated protocol. MicroRNA expression was evaluated using a specific TaqMan MicroRNA Assay as described by the manufacturer (Applied Biosystems, Waltham, MA, USA). Briefly, 10 ng of RNA wasreversetranscribed using the TaqMan MicroRNA Reverse Transcription Kit primed with the specific RT primer. Real-time PCR was performed in triplicate using specific primers. The expression of each miRNA was normalized to the RNU44 expression. To evaluate the expression of the selected genes, 100 ng of RNA wasreversetranscribed using the SuperScript VILO IV cDNA Synthesis Kit (Invitrogen, Waltham, MA, USA). The cDNA was used for real-time PCR using the specific primers contained in the TaqMan Gene Expression Assay (Applied Biosystems). Gene expression was normalized to the GAPDH expression. Experiments were performed in triplicate.

### 4.4. Statistical Analysis

Student’s *t*-test (two-tailed test) was used to compare mean values between couples of samples. Graphic representation and statistical analysis have been performed using GraphPad Prism 6.0c (GraphPad Software).

## Figures and Tables

**Figure 1 genes-15-00820-f001:**
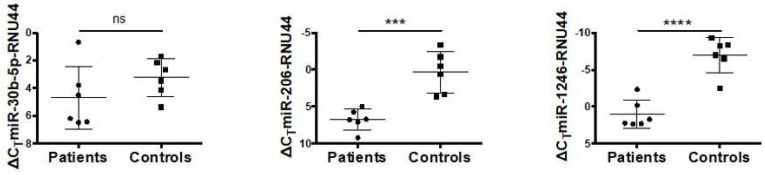
MiR-1246, miR-30b, and miR-206 expression in bone marrow hematopoietic cells from FA patients and healthy donors. RNU44 was used as a reference control. Data (mean ± SD) are from an experiment performed in triplicate. Statistical analysis was performed by Student’s *t*-test. *** *p* < 0.001, **** *p* < 0.0001, ns: not significant.

**Figure 2 genes-15-00820-f002:**
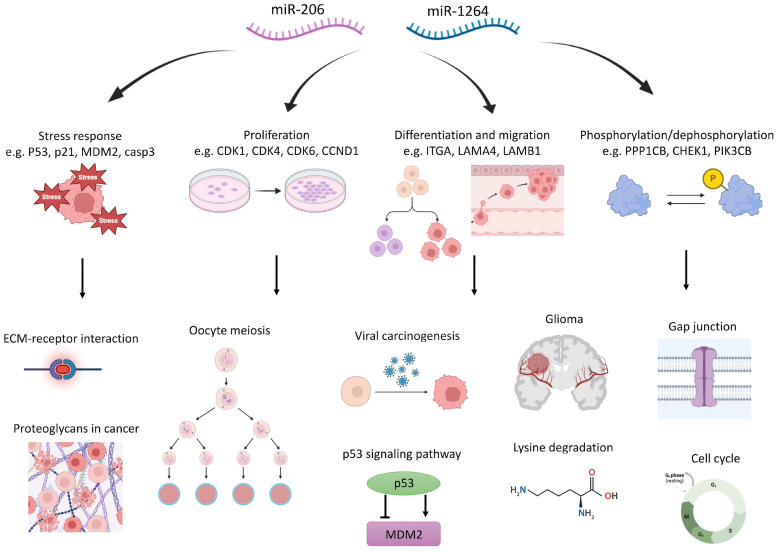
A schematic representation of the biochemical and functional pathways modulated by miR-206 and miR-1264.

**Figure 3 genes-15-00820-f003:**
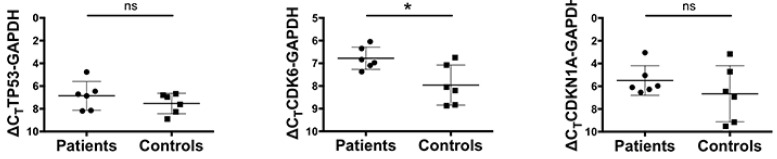
Expression of TP53, CDK6, and p21 in bone marrow hematopoietic cells from FA patients and healthy donors. GAPDH was used as a reference control. Data (mean ± SD) are from an experiment performed in triplicate. Statistical analysis was performed by Student’s *t*-test. * *p* < 0.05, ns: not significant.

**Figure 4 genes-15-00820-f004:**
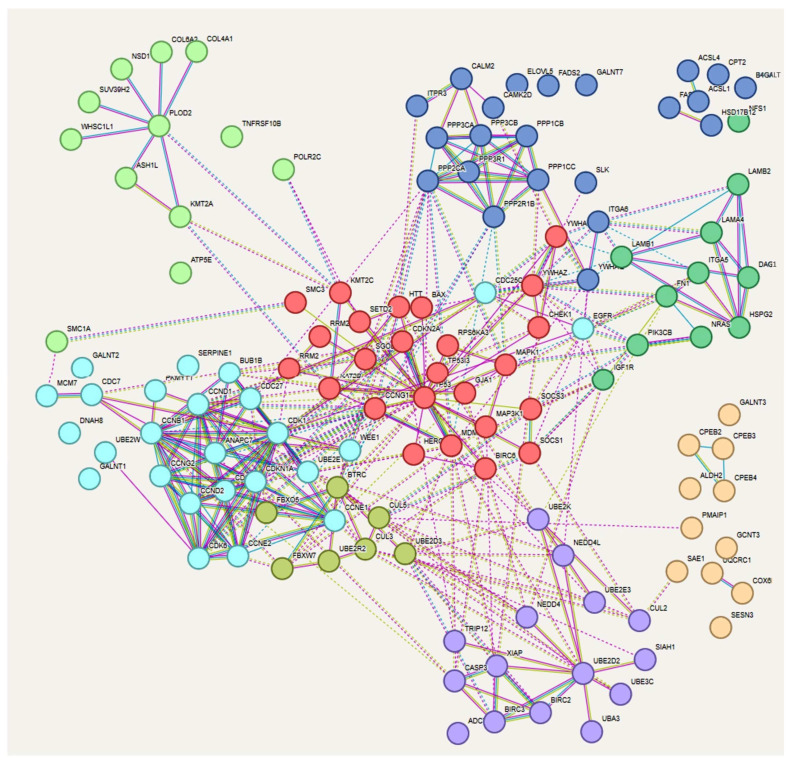
Visualization of the complex network using the open-source software platform STRING functional protein association networks (https://string-db.org/ accessed on 15 June 2023)) and Cytoscape (https://cytoscape.org/ accessed on 15 June 2023)). Pathway-related gene clouds are differently colored: Cyan and Red clouds (in the center) is for Pathway 7, blue clouds are for Pathway 1, 3, 12 (top right) and Pathway 4 (top center), the lime green cloud for Pathway 5 (top left), the dark green cloud (on the right) is for Pathways 5 and 15, the violet cloud (bottom center) is for Pathway 6,and the pink cloud (bottom right) is for Pathway 2.

**Table 1 genes-15-00820-t001:** List of the microRNAs selected by the enrichment procedures. The trend in expression is also reported. Bold indicates the miRNAs validated by real-time PCR.

MicroRNA	Up- (U) or Downregulated (D)
hsa-miR-181b-3p	U
**hsa-miR-1246**	**D**
hsa-miR-129-1-3p	U
**hsa-miR-206**	**D**
hsa-miR-23b-5p	D
**hsa-miR-30b-5p**	**D**

**Table 2 genes-15-00820-t002:** Pathways identified after enrichment plus validation processes of miRNAs associated with FA sample from Table 1. The table reports the number of genes, the miRNAs associated with each pathway, and the statistical parameter attributed to the pathways with a relationship to the respective associated miRNA.

#	KEGG Pathway	*p*-Value	#Genes	#GENES/MIR	#miRNAs	Adj *p*-Value
1.	Fatty acid biosynthesis (hsa00061)	<1 × 10^−325^	3	3	hsa-miR-30b-5p	8.210 × 10^−11^
2.	Mucin-type O-glycan biosynthesis (hsa00512)	<1 × 10^−325^	6	6	hsa-miR-30b-5p	7.029 × 10^−14^
3.	Fatty acid metabolism (hsa01212)	<1 × 10^−325^	7	7	hsa-miR-30b-5p	4.032 × 10^−21^
**4.**	**Oocyte meiosis (hsa04114)**	<1 × 10^−325^	33	8	**hsa-miR-1246**	3.066 × 10^−6^
				4	**hsa-miR-206**	5.98 × 10^−3^
				25	hsa-miR-30b-5p	8.210 × 10^−23^
**5.**	**ECM-receptor interaction (hsa04512)**	<1 × 10^−325^	10	9	hsa-miR-23b-5p	3.506 × 10^−55^
				2	hsa-miR-129-1-3p	6.747 × 10^−12^
				1	**hsa-miR-206**	2.93 × 10^−4^
6.	Ubiquitin-mediated proteolysis (hsa04120)	5.62 × 10^−12^	30	30	hsa-miR-30b-5p	3.339 × 10^−6^
**7.**	**p53 signaling pathway (hsa04115)**	2.90 × 10^−11^	23	4	hsa-miR-181b-3p	6.33 × 10^−3^
				6	**hsa-miR-1246**	1.49 × 10^−3^
				15	hsa-miR-30b-5p	1.230 × 10^−10^
**8.**	**Lysine degradation (hsa00310)**	3.99 × 10^−10^	9	2	**hsa-miR-1246**	6.53 × 10^−3^
				9	hsa-miR-30b-5p	1.246 × 10^−2^
**9.**	**Viral carcinogenesis (hsa05203)**	1.31 × 10^−8^	6	6	**hsa-miR-1246**	4.920 × 10^−11^
**10.**	**Proteoglycans in cancer (hsa05205)**	1.52 × 10^−7^	10	4	hsa-miR-181b-3p	4.420 × 10^−6^
				4	hsa-miR-23b-5p	4.27 × 10^−4^
				2	hsa-miR-129-1-3p	4.00 × 10^−3^
				3	**hsa-miR-206**	2.35 × 10^−2^
**11.**	**Cell cycle (hsa04110)**	5.99 × 10^−7^	21	9	**hsa-miR-206**	7.700 × 10^−7^
				12	hsa-miR-30b-5p	4.37 × 10^−4^
12.	Biosynthesis of unsaturated fatty acids (hsa01040)	8.14 × 10^−7^	3	1	hsa-miR-23b-5p	2.16 × 10^−2^
**13.**	**Glioma (hsa05214)**	3.88 × 10^−6^	9	5	hsa-miR-181b-3p	3.12 × 10^−4^
				4	**hsa-miR-1246**	7.76 × 10^−4^
**14.**	**Gap junction (hsa04540)**	1.69 × 10^−4^	3	3	**hsa-miR-206**	1.310 × 10^−8^
15.	Small-cell lung cancer (hsa05222)	2.56 × 10^−4^	12	6	hsa-miR-23b-5p	8.250 × 10^−6^
				7	hsa-miR-30b-5p	2.12 × 10^−2^
16.	Apoptosis (hsa04210)	2.18 × 10^−3^	8	8	hsa-miR-30b-5p	2.080 × 10^−3^
17.	Chronic myeloid leukemia (hsa05220)	3.22 × 10^−3^	8	5	hsa-miR-181b-3p	1.52 × 10^−4^

## Data Availability

No new data were generated for the analysis presented in this paper. Original data were previously reported ([1]: Degan P. et al., Metab Syndr Relat. Disord. 2019, 1, 53-59). The software employed here for their analysis is freely available online.

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
