# Peer review of "Advanced Analysis and Validation of a microRNA Signature for Fanconi Anemia"

_genes, 2024, doi:10.3390/genes15070820_

Round 1

Reviewer 1 Report

Comments and Suggestions for Authors

This work is a follow-up of the previous study (Degan et al., 2019), and it comes up with a bioinformatics-based strategy to find potential biomarkers for Fanconi anemia. The authors analyze specific miRNA signature from six healthy individuals and six Fanconi anemia patients. Out of the six potential miRNA candidates (miRNA-129-1-3p, miR-107 NA181b-3p, miRNA-23b-p5, miRNA-30b-5p, miRNA-206 and miRNA-1246) obtained in the in silico analyses, they validated three of the miRNAs by RT-PCR in the bone marrow mononuclear cells. miRNA-206 and miRNA-1246 shows a significant reduction in the cells from Fanconi anemia patients. Since some of the miRNA regulate p53 signaling, a central player in Fanconi anemia, the authors further evaluated the expression levels of p53 and other downstream target gene expression levels.  CDK6 levels were upregulated significantly in Fanconi anemia patient samples. Altogether, they propose that miRNA-206 could be a central player regulating signaling events in the Fanconi anemia patients. Overall, the study is designed and executed nicely.
Here are my suggestions:

   1.  Fanconi anemia is written as FA and FANCA. The authors should follow the same nomenclature throughout the manuscript.

   2. It will be easier to comprehend the data if the author could show the fold changes instead of relative delta CT values for the RT-PCR results.

Author Response

Reviewer 1

This work is a follow-up of the previous study (Degan et al., 2019), and it comes up with a bioinformatics-based strategy to find potential biomarkers for Fanconi anemia. The authors analyze specific miRNA signature from six healthy individuals and six Fanconi anemia patients. Out of the six potential miRNA candidates (miRNA-129-1-3p, miR-107 NA181b-3p, miRNA-23b-p5, miRNA-30b-5p, miRNA-206 and miRNA-1246) obtained in the in silico analyses, they validated three of the miRNAs by RT-PCR in the bone marrow mononuclear cells. miRNA-206 and miRNA-1246 shows a significant reduction in the cells from Fanconi anemia patients. Since some of the miRNA regulate p53 signaling, a central player in Fanconi anemia, the authors further evaluated the expression levels of p53 and other downstream target gene expression levels.  CDK6 levels were upregulated significantly in Fanconi anemia patient samples. Altogether, they propose that miRNA-206 could be a central player regulating signaling events in the Fanconi anemia patients. Overall, the study is designed and executed nicely.
Here are my suggestions:

   1.  Fanconi anemia is written as FA and FANCA. The authors should follow the same nomenclature throughout the manuscript.

As requested by the reviewer we have corrected the nomenclature by using FA to indicate Fanconi anaemia.

  1. It will be easier to comprehend the data if the author could show the fold changes instead of relative delta CT values for the RT-PCR results.

We thank the reviewer for this suggestion. However, we left the graph in delta CT because reporting expression data in terms of fold change would result in an excessively dispersed distribution of points corresponding to single samples. Instead, delta CT representation enables a discrete grouping of points useful for an immediate view of the relative distribution of expression data between patients and controls.

Reviewer 2 Report

Comments and Suggestions for Authors

The brief report by Cappelli and colleagues focuses on the characterization of 2 micro RNAs on Fanconi Anemia biology. This study stems from a previous publication by the same group and opens up potential future validations. The current work is mostly exploratory and selective, while a more generalized and genome-wide approach would have increased the interest of the report. In my opinion, the effort provided by the authors is just setting up the ground for future research, while this work itself has limited potential. Aside from this general consideration, the paper is well-written and offers some curious insight. However, some aspects need to be addressed to increase the quality and readability of the report.

-              The authors should provide more details about the bioinformatic pipeline that led them to select  6 out of 24 initial FA-associated miRNAs. This is a key element of the report and, although some aspects are described in the methods, it is necessary to provide more detailed information in the main text.

-              Please add the statistical test used (with P-value) to the legend of both experimental figures.

-              In lines 104-107 the authors seem to cherry-pick their conclusions... The sample number is scarce (justified by the rarity of the disease model), but they shouldn’t comment on it as favorable when the result is significant and unfavorable when it is not. I would remove any unnecessary comment; statistical results are objective.

-              While in the “results” section the authors have extensively commented on the role of miR-206 and miR-1246, in the “conclusions” section they wrap-up their concepts only for miR-206. Please expand your conclusions

-              A conclusive scheme/figure on the pathways emerging as impacted by miR-206 and miR-1246 in FA would help the reader following the flow of reasoning.

Author Response

Reviewer 2

The brief report by Cappelli and colleagues focuses on the characterization of 2 micro RNAs on Fanconi Anemia biology. This study stems from a previous publication by the same group and opens up potential future validations. The current work is mostly exploratory and selective, while a more generalized and genome-wide approach would have increased the interest of the report. In my opinion, the effort provided by the authors is just setting up the ground for future research, while this work itself has limited potential. Aside from this general consideration, the paper is well-written and offers some curious insight. However, some aspects need to be addressed to increase the quality and readability of the report.

-              The authors should provide more details about the bioinformatic pipeline that led them to select  6 out of 24 initial FA-associated miRNAs. This is a key element of the report and, although some aspects are described in the methods, it is necessary to provide more detailed information in the main text.

As suggested by the reviewer, we add more details about the bioinformatic tools used.

“Since miRNAs regulation is a degenerate functional relationship where one miRNA targets multiple genes and each target gene can be simultaneously targeted by more than one miRNA, it has become indispensable to elaborate in silico tools able to reasonably shorten the list of miRNA-target interactions.This helps to better focus onthe regulatory mechanism orchestrated by different miRNAs in the various cellular processes so to prioritize miRNA-target interactions and maximize the effectiveness of the downstream analysis. To date, no straightforward process has been devised to assistin data analysis procedures. For example, in our original report [1,] the 24 miRNAs resulting from the analysis were distributed in upregulated (9) and downregulated (15) miRNAs, respectively associated with 31 and 30 pathways. Among these, only 14 upregulated and 13 downregulated pathways were independent. The two miRNA populations influence two different gene populations, which overlap only partially. This problem poses different questions regarding the meaning of this data redundancy as well as which procedure might be the most appropriate to adopt for the analysis. There is not even much literature dedicated to discussing what the preferential approaches might be.However, several sophisticated analysis tools have luckily been developed recently, addressing and overcoming several problems. Among the various analysis tools freely available on the web, we found that MIENTURNET was particularly suited for our purposes due to its versatility and ease of use.MIENTURNET is a user-friendly web tool that integrates the miRNA-target enrichment analysis with a network-based visualization and analysis, addressing the prioritization problem of miRNA-target candidate interactions. The procedure embedded in MIENTURNET assumes basic skills in network theory, and statistics and the software does not require bioinformatics expertise.With MIENTURNET, a list of mature miRNAs or genes can be analyzedto infer all possible computational and experimental evidence of their regulation on target genes, or miRNAs, based on a statistical overrepresentation analysis of their specific interactions. These interactions are consequently visualized as a network and analyzed according to their topological features to gain insights into understanding the biological processes underlying the target gene activity, by acquiring information from different data platforms generated by the analysis: KEGG, Reactome, WikiPathways, and Disease Ontology.Theoriginal population of 24 miRNA associated with the analyzed FA samples [1], selected through MIENTURNET, yielded a panel of 6 miRNAs (Table 1).These miRNAs are associated with the 17 pathways (Table 2). Table 2 reports data consistent with KEGG MIENTURNET analysis. Alternatively, the software also generates information for Reactome, WikiPathways, and Disease Ontology (Supplementary Fig. 1).”

-              Please add the statistical test used (with P-value) to the legend of both experimental figures.

As suggested by the reviewer, we have added statistical test used and P-value to the legend of both experimental figures.

-              In lines 104-107 the authors seem to cherry-pick their conclusions... The sample number is scarce (justified by the rarity of the disease model), but they shouldn’t comment on it as favorable when the result is significant and unfavorable when it is not. I would remove any unnecessary comment; statistical results are objective.

We agree with the reviewer and, as rightly suggested, we have removed the sentence in question.

-              While in the “results” section the authors have extensively commented on the role of miR-206 and miR-1246, in the “conclusions” section they wrap-up their concepts only for miR-206. Please expand your conclusions

As rightly suggested by the reviewer, we expand the conclusion underlying the role of miR-1246 in FA cells.

It is however necessary to underline the role of miR-1246 as an important partner in shaping the pathophysiological manifestations of FA. Indeed miR-1246 is involved in the defective p53 pathway, in the signaling through cell cycle progression by affecting the balance between phosphorylation/dephosphorylation and cyclins and kinase activities and in the redox unbalance.”

-              A conclusive scheme/figure on the pathways emerging as impacted by miR-206 and miR-1246 in FA would help the reader following the flow of reasoning.

As suggested by the reviewer, we have added a figure showing a schematic representation of the biochemical and functional pathways played by these miRNAs (New Figure 3).